# Ferroptosis in Cardiovascular Diseases: Current Status, Challenges, and Future Perspectives

**DOI:** 10.3390/biom12030390

**Published:** 2022-03-02

**Authors:** Yi Guo, Chanjun Lu, Ke Hu, Chuanqi Cai, Weici Wang

**Affiliations:** 1Clinic Center of Human Gene Research, Union Hospital, Tongji Medical College, Huazhong University of Science and Technology, Wuhan 430022, China; hxshuyun@gmail.com; 2Department of Vascular Surgery, Union Hospital, Tongji Medical College, Huazhong University of Science and Technology, Wuhan 430022, China; m202075945@hust.edu.cn (C.L.); 2020508102@hust.edu.cn (K.H.); chuanqicai@hust.edu.cn (C.C.)

**Keywords:** ferroptosis, cardiovascular diseases, ischemia/reperfusion injury, heart failure, cardiomyopathy, atherosclerosis

## Abstract

Cardiovascular diseases (CVDs) are still a major cause of global mortality and disability, seriously affecting people’s lives. Due to the severity and complexity of these diseases, it is important to find new regulatory mechanisms to treat CVDs. Ferroptosis is a new kind of regulatory cell death currently being investigated. Increasing evidence showed that ferroptosis plays an important role in CVDs, such as in ischemia/reperfusion injury, heart failure, cardiomyopathy, and atherosclerosis. Protecting against CVDs by targeting ferroptosis is a promising approach; therefore, in this review, we summarized the latest regulatory mechanism of ferroptosis and the current studies related to each CVD, followed by critical perspectives on the ferroptotic treatment of CVDs and the future direction of this intriguing biology.

## 1. Introduction

Ferroptosis, a new kind of cell death, originates from the Latin word “ferrum” and the Greek word “ptosis”, meaning “iron” and “a fall”, respectively [1]. Dixon and his colleagues first reported it in 2012 as an iron-dependent form of nonapoptotic cell death [2]. It is characterized by dense, compact mitochondria with a loss of crista, which is unique and distinct from other forms of cell deaths, such as apoptosis (chromatin margination and condensation), necrosis (ruptured plasma membrane and swollen cytoplasm), and autophagy (enclosed double-membrane vesicles formation) [3,4]. Ferroptosis was determined to participate in many diseases such as tumors, neurodegenerative disorders (Alzheimer’s disease, Parkinson’s disease, Huntington’s disease etc.), stroke, and ischemia/reperfusion (I/R), and it became a hotspot and focus of research, meriting further exploration.

Cardiovascular diseases (CVDs), a major cause of global mortality and disability, greatly affect people’s lives [5]. One ongoing multinational collaboration study shows that the burden of CVDs was increasing in nearly all countries [6]. The age-standardized rate of CVDs is even increasing in some high-income countries that previously had a declining rate [7]. Recent research demonstrated that ferroptosis participates in several CVDs, such as I/R [8,9,10,11], heart failure (HF) [12,13], cardiomyopathy [14,15], and atherosclerosis [16]. Given the huge potential of ferroptosis in the treatment of CVDs, the latest work should be summarized, together with a report on the progress of possible medical usage.

In this review, we present the latest regulatory mechanism of ferroptosis and current studies related to each CVD, followed by critical perspectives on present challenges on the ferroptotic treatment of CVDs and the prospects of this intriguing biology.

## 2. The Regulating Pathways of Ferroptosis

Ferroptosis was first proposed by Dixon as an iron-dependent regulated cell death [17]. Triggered by small specific molecules such as erastin, a small RAS gene selective molecule, and RAS-selective lethal 3 (RSL3), ferroptosis is characterized by the unique morphological changes of mitochondria. The overproduction of lipid peroxides, not clearing in time, increases membrane permeability, reduces fluidity, and destroys ionic homeostasis; thus, affecting the normal structure and function of membrane, ultimately leading to cell death [18]. Iron metabolism is critical in the formation of lipid peroxides [19]. Xc-GSH-GPX4 pathway [20], FSP1–CoQ10–NAD(P)H pathway [21,22], GCH1–BH4–DHFR pathway [23,24], and mitochondria [25] all play an important role in scavenging excess lipid peroxides and regulating the process of ferroptosis (Figure 1).

### 2.1. Iron Metabolism

Iron overload can lead to ferroptosis. Iron, the most abundant trace element in vivo, is regulated to maintain appropriate levels. Apart from being part of the composition of the human body, iron can be detrimental due to its active redox capacity, such as Fenton and Fenton-like reactions, which means ferrous iron (Fe2+) can react with oxygen or hydrogen peroxide (H_2_O_2_) and produce plenty of hydroxyl radicals and lipid peroxides, thus ultimately leading to ferroptosis [19]; therefore, regulating iron homeostasis is of great importance. Iron homeostasis is related to nuclear receptor coactivator 4 (NCOA4)-mediated ferritinophagy, a kind of autophagy that turns ferritin into intracellular iron [26,27,28,29]. Heme oxygenase-1 (Hmox1) also contributes to iron homeostasis, which exerts the degradation of heme and produces ferrous iron [30,31]. The activation of ferritinophagy or overexpression of Hmox1can increases the free iron level, leading to lipid peroxides accumulation and ultimately ferroptosis. In addition, BTB domain and CNC homolog 1 (BACH1), a transcription factor, can promote ferroptosis by downregulating ferritin genes, including ferritin heavy chain 1 (Fth1) and ferritin light chain 1 (Ftl1), which reduce the amount of free labile iron. Therefore, it is possible to inhibit BACH1 to reduce free labile iron and further alleviate ferroptosis [32].

### 2.2. Lipid Peroxidation

Ferroptosis can be induced by lipid peroxidation, consisting of nonenzymatic lipid peroxidation like Fenton reaction and enzymatic lipid peroxidation mediated by the activity of the lipoxygenase (LOX) family. Poly-unsaturated fatty acid-phosphatidyl ethanolamine (PUFA-PE) can be metabolized by Fe2+ and LOX to produce lipid peroxides. Poly-unsaturated fatty acid (PUFA), especially arachidonic acid (AA) and adrenaline, can be catalyzed by acyl-CoA synthetase long-chain family member 4 (ACSL4) and lysophosphatidylcholine acyltransferase 3 (LPCAT3), and they then take part in the biosynthesis of PUFA-PE with phosphatidylethanolamine (PE). Once the generation of lipid peroxides is hyperactive, lasting depletion of PUFA affects the normal structure and function of membrane and eventually leads to cell death [19]. ACSL4 is identified as a pivotal biomarker of ferroptosis. Knocking down ACSL4 by shRNA inhibits ferroptosis, whereas the overexpression of ASCL4 contributes to the sensitivity of ferroptosis by modulating the cellular lipid composition [33]. Special protein 1 (Sp1), an important factor, activates ACSL4 transcription when binding to the promoter region of ACSL4 [34]; consequently, increasing the expression or catalytic activity of ASCL4, LPCAT3, LOX, or Fenton reaction leads to lipid peroxides accumulation and, ultimately, ferroptosis [35].

### 2.3. Xc-GSH-GPX4 Pathway

The Xc-GSH-GPX4 pathway is the main pathway to regulate ferroptosis. Ferroptosis is mainly induced by a large number of reactive oxygen species (ROS), far beyond the capability of clearing mechanisms. The main clearing mechanism that reduces ROS is the redox ability of glutathione (GSH). GSH, a tripeptide that includes glutamic acid, cysteine, and glycine, acts as an antioxidant and is the substrate of glutathione peroxidase 4 (GPX4), which is then converted into oxidized glutathione (GSSG). Increasing the expression of GSH can inhibit ferroptosis. Furthermore, the System Xc−, composed of subunit solute carrier family 7 member 11 (SLC7A11) and solute carrier family 3 member 2 (SLC3A2), mediates the exchange of cystine and glutamate across the plasma membrane at the ratio of 1:1 to synthesize GSH. The inhibition of System Xc− by erastin can lead to ferroptosis due to GSH depletion. This inhibition may cause the complete opposite effect of inhibiting ferroptosis. Inhibiting SLC7A11 decreases the amount of cystine and causes NADPH “debt” to reduce cystine to cysteine, thus impairing antioxidant ability and driving cells to ferroptosis [36]. As for GPX4, it is a kind of selenoenzyme that reduces ROS through GSH [20]. The inhibition of GPX4 by RSL3 leads to an impairment of antioxidant capacity, and then to ferroptosis.

Moreover, P53 can also regulate ferroptosis in two diametrically opposed ways [37]. On the one hand, some studies report that P53 can lower the expression of SLC7A11, which affects the cystine transport function of system Xc- and further inhibits the activity of GPX4, ultimately leading to ferroptosis [38]. On the other hand, P53 is reported to inhibit ferroptosis in some cells, involving cyclin-dependent kinase inhibitor 1 A (CDKN1A), a p53 transcriptional target [39].

Furthermore, nuclear factor erythroid 2-related factor 2 (Nrf2) increases the level of SLC7A11 and transcriptionally induces the expression of GPX4, which then reduces ROS; therefore, the overexpression of Nrf2 can inhibit ferroptosis, whereas the Kelch-like ECH associated protein 1 (Keap1), bonding to Nrf2 and negatively regulating it, can reverse this process and exert a proferroptosis role [40,41]. However, Nrf2 can also induce ferroptosis by upregulating Hmox1 then degrading heme and releasing free iron, which ultimately leads to ferroptosis [14].

Cystine starvation can lead to ferroptosis because cysteine is crucial to GSH synthesis as a rate-limiting substrate, and the followed glutamate accumulation increases ROS. Interestingly, in nonsmall-cell lung cancer (NSCLC) cells, cystine starvation inducing ferroptosis can be inhibited by the generation of γ-glutamyl-peptide for reducing glutamate stress, which is catalyzed by glutamate–cysteine ligase catalytic subunit (GCLC), a Nrf2 regulating protein [42]. In high-fat diet-induced nonalcoholic fatty liver disease (NAFLD), ferroptosis contributes its development, and ginkgolide B (GB) is effective for treatment by inhibiting lipid accumulation and oxidative stress, which are probably related to Nrf2 elevation induced by GB [43]. All these show that Nrf2 is a crucial target regarding ferroptosis.

### 2.4. FSP1–CoQ10–NAD(P)H Pathway

Ferroptosis suppressor protein 1 (FSP1), previously named apoptosis-inducing factor mitochondria-associated 2 (AIFM2), is thought to inhibit ferroptosis as an independent system that co-operates with the Xc-GSH-GPX4 pathway [21,22]. That means FSP1 is capable to eliminate lethal lipid peroxides even in the absence of GPX4. Acting as an oxidoreductase, FSP1 reduces ubiquinone (also named coenzyme Q10) into ubiquinol by NADPH, another antioxidant compound besides GSH, which can inhibit ferroptosis by lessening lipid ROS. Moreover, inhibiting the mevalonate pathway may decrease the amount of coenzyme Q10, thus leading to ferroptosis [44] However, a study also showed that ubiquinol production is not essential for ferroptosis resistance mediated by FSP1, while endosomal sorting complexes required for transport III(ESCRT-III)–dependent membrane repair is responsible for it [45]. Anyway, they all elaborate that FSP1 is a potential target for ferroptosis-related diseases.

### 2.5. Mitochondria

The dramatic morphological features of ferroptosis are the dense, compact mitochondria with the loss of crista. It is unique and distinct from other forms of cell death, such as apoptosis (chromatin margination and condensation), necrosis (ruptured plasma membrane and swollen cytoplasmic), and autophagy (enclosed double-membrane vesicles formation) [46]. Voltage-dependent anion channels (VDACs), the ample protein located in the outer mitochondrial membrane, are essential to ferroptosis as a potential erastin target. The more VDACs proteins cells exist, the more sensitivity these cells exhibit to ferroptosis [47]. Moreover, the mechanism of mitochondria participating in ferroptosis involves cystine starvation and glutaminolysis, which raise the content of membrane potential (MMP) and ferrous iron content and contribute to Fenton reaction and lipid peroxidation, leading to ferroptosis [48,49].

Dihydroorotate dehydrogenase (DHODH), found on the inner membrane of mitochondrial, is reported to generate ubiquinol from ubiquinone by modulating the transformation of dihydroorotate to orotate, which demonstrates as a parallel way from mitochondrial GPX4 to protect cells from lipid peroxidation. Moreover, GPX4 knocks down sensitizing human tumor cells to DHODH inhibition. DHODH inhibitor maybe the potent anticancer agent in low GPX4 expression cancer. This mechanism of DHODH in mitochondrial is similar to the FSP1 system in cytosolic. Overexpression of FSP1 on mitochondria cannot affect lipid peroxidation, suggesting that the ferroptosis regulating the function of FSP1 may require some proteins on the plasma membrane [25].

### 2.6. GCH1–BH4–DHFR Pathway

The guanosine triphosphate cyclohydrolase 1 (GCH1)–tetrahydrobiopterin (BH4)–dihydrofolate reductase (DHFR) pathway is another regulatory mechanism independently from Xc-GSH-GPX4 axis. Under the induction of ferroptosis by RSL3, erastin, and Gpx4 knockout, GCH1 exerts strong protective effect, which is selective from other forms of cell death [23]. The upregulation of GCH1 increases the content of BH4, an endogenous antioxidant, which captures lipid-derived peroxyl radicals to block the process of lipid peroxidation. Moreover, DHFR is the most efficient way for the regeneration of BH4 [24]; therefore, GCH1–BH4–DHFR pathway inhibits ferroptosis by subverting lipid peroxidation.

### 2.7. Energy Stress

Energy stress is one of metabolic stress, characterized by the increase in intracellular AMP and the consumption of intracellular ATP. Adenosine monophosphate (AMP)-activated protein kinase (AMPK) is crucial to energy stress by stimulating the ATP-producing catabolic processes and inhibits ATP-consuming anabolic processes [50]. Activation of AMPK in energy stress increases phosphorylation of acetyl-CoA carboxylase (ACC), thus hindering the synthesis and oxidation of fatty acids, which illustrates how energy stress inhibits ferroptosis [51]. A study also reports that AMPK promotes ferroptosis by phosphorylating BECN1 (beclin 1), binding to SLC7A11, then inhibiting cystine transport [52]; therefore, the precise effect of energy stress or AMPK needs further investigation.

## 3. Ferroptosis in Different Cardiovascular Cells

### 3.1. Endothelial Cells

It is suggested that erastin can induce ferroptosis in human umbilical vein endothelial cells (HUVECs), and the overexpression of miRNA17-92 can greatly alleviate it through A20 (known as tumor necrosis factor alpha-induced protein 3, TNFAIP3)-ACSL4 axis [1].When exposed to PM2.5, endothelial cells are injured due to ferroptosis because of increased iron content and lipid overoxidation [2]. In addition, overloaded iron induces ferroptosis and apoptosis to promote the calcification of endothelium, which can be alleviated by the application of ferroptosis inhibitors and iron chelators [53]. One study showed that high systemic iron levels can lead to the activation and dysfunction of the endothelium, which ultimately promote atherosclerosis formation [54]. Ferroptosis inhibitors like ferrostatin-1 and deferoxamine mesylate can partially rescue this kind of endothelial lesion [2]. In subarachnoid hemorrhage (SAH) model, treating endothelial cells with cepharanthine (CEP) exhibits less 15-lipoxygenase-1 (ALOX15) level and less typical ferroptotic morphological changes in mitochondria [55]. Reported to promote vascular regeneration and improve endothelial function, Astragaloside IV is helpful to protect endothelial cells from ferroptosis by inhibiting lysophosphatidylcholine (LPC)-induced ROS in HUVECs, which can be largely reversed by ferroptosis agonist. All these studies suggest that ferroptosis is a promising target to protect endothelium [56].

### 3.2. Vascular Smooth Muscle Cells

Interestingly, one study shows that cigarette smoke extract (CSE) can induce ferroptosis by increasing lipid peroxidation and decreasing intracellular GSH in A7r5 cells and primary rat vascular smooth muscle cells (VSMCs), which can be completely inhibited by ferrostatin-1, liproxstatin-1, and deferoxamine—the specific ferroptosis inhibitors. Acrolein and methyl vinyl ketone in CSE were determined to be the main causes [3]. Treating VSMCs with palmitic acid (PA) upregulates the expression of extracellular matrix protein periostin (POSTN), which inhibits SLC7A11 expression by suppresses the expression of P53 gene in VSMCs, and therefore reduces the synthesis of GSH, leading to ferroptosis. However, pretreating with metformin activates Nrf2 pathway, a powerful antioxidant system, significantly reducing ferroptosis in VSMCs [57]. All of these indicate that we can target ferroptosis to protect VSMCs.

### 3.3. Macrophages

As for ferroptosis in macrophages, one study shows that iron load, which is a contributor to ferroptosis, can trigger the macrophage’s polarization toward a proinflammatory phenotype, M1 [58]. M1 macrophages, with strong expression of inducible nitric oxide synthase (iNOS)/NO•, highly resist RSL3, whereas M2 cells lacking iNOS/NO• are sensitive to it; therefore, iNOS/NO• has antiferroptosis effects and is the reason why M1 macrophages can exist with great viability under excess iron [4]. Macrophage migration inhibitory factor (MIF), overexpressed in nasopharyngeal carcinoma (NPC) cells, significantly inhibits ferroptosis in macrophages, which is conducive to NPC metastasis [59]. Treating macrophage with LPS to generate sepsis models in vitro, it is shown that both 4-octyl itaconate (4-OI) and ferrostatin-1 significantly upregulate the expression of GPX4, SLC7A11C, and Nrf2, exhibiting a protective effect for THP-1 macrophage and inhibiting lipid peroxidation induced by LPS. Further, this ferroptosis resistance is highly dependent on Nrf2 [60].

### 3.4. Cardiomyocytes

Ferroptosis-related cardiomyocyte death was reported in several cardiovascular diseases. Owing to the stimuli of an abundance of free radicals, cardiomyocytes suffer from ferroptosis and cannot be regenerated; therefore, it is crucial to inhibit ferroptosis, which can maintain the structure and function of the heart [8].

Ferroptosis is related to doxorubicin (DOX)-induced cardiotoxicity through upregulating Hmox1, which can release iron by degrading heme in cardiomyocytes, leading to plenty of cardiotoxic reactive oxygen species (ROS) [15]. Moreover, diabetes creates a high risk of cardiovascular disease, and that inhibition of ferroptosis can reduce cardiomyocyte injury in high glucose models, which relates to endoplasmic reticulum stress (ERS) aggravation [9,61]. It was demonstrated that the mechanistic target of rapamycin (mTOR) is protective in cardiomyocytes treated with ferric citrate, erastin, or RSL3, which can all induce ferroptosis. The protective role of mTOR is due to the reduction in excess iron and suppression of ROS [62]. One study reported that overexpressing Slc7a11 and treatment with Fer-1 can protect against ferroptosis in cardiomyocytes [63]. A recent study shows that the expression of USP22 protein, a kind of deubiquitinase, can inhibit ferroptotic cardiomyocyte death through the SIRT1/p53/SLC7A11 axis and further protect against myocardial ischemia/reperfusion injury [64].

In the ongoing COVID-19 epidemic, one case report shows that ferroptosis signature, immunohistochemical staining with E06, which reflects lipid peroxidation, is positive in COVID-19 myocardium and is negative in unknown-etiology viral myocarditis. The result may indicate that COVID-19 myocardium is related to ferroptosis, which merits further exploration [65].

## 4. Ferroptosis in Cardiovascular Diseases

### 4.1. Ferroptosis in Ischemia/Reperfusion (I/R)

Inhibiting ferroptosis can reduce I/R injury. I/R and ferroptosis were studied in several organs, such as the heart [8,9,10,11,29], kidney [51,66,67,68], brain [69], intestine [8], and liver [70] (Table 1). I/R includes two parts: ischemia and reperfusion. Ischemia means the sudden block of blood supply in an aerobic organ, and reperfusion means the restoration of blood supply. These processes lead to excessive production of ROS and free radicals, which result in the inflammatory cascade, cell injury, and ferroptosis [71].

ROS are central to I/R. Inhibiting ERS, triggered by ROS, can alleviate ferroptosis in diabetes myocardial I/R injury [9]. Compared with nondiabetic hearts, the expression of GPX4 decreases while malondialdehyde (MDA) and 4-hydroxynonenal (4-HNE) increases during myocardial I/R injury in diabetic rats. NADPH oxidase (Nox), the main donor of ROS, is responsible for the oxidative stress in diabetic rat hearts, which is closely related to AMPK [75]. It is reported that the inhibition of glutaminolysis reduces the amount of ROS and increases GSH, thus inhibiting ferroptosis in heart I/R injury [76]. Augmenter of liver regeneration (ALR), colocalized with GPX4, possesses the ability to inhibit ferroptosis by decreasing ROS [67].

Other pathways elucidate the role of ferroptosis in I/R. Liproxstatin-1 (Lip-1), a ferroptosis inhibitor, protects heart I/R injury by reducing voltage-dependent anion channel 1 (VDAC1) levels and increasing GPX4 levels [10,11]. Iron homeostasis is also important because a cohort study shows that the ability to rapidly process iron will be impaired during kidney I/R, which leads to ferroptosis in the end [68].

Ferroptosis can also promote the neutrophil’s adhesion to coronary vascular endothelial cells by Toll-like receptors 4 (TLR4)/TIR domain-containing adapter inducing IFN-beta (Trif)/type I IFN signaling pathway, which aggravates inflammation and leads to I/R injury [8]. Additionally, overexpression of miR-214 or silencing plasmacytoma variant 1 (PVT1) could significantly inhibit ferroptosis and reduce brain infarct size in mice [69]. Renal I/R injury can be attenuated by pannexin 1 (PANX1) depletion for a decrease in mitogen-activated protein kinase (MAPK)/extracellular signal-regulated kinase (ERK) activation, which inhibits ferroptosis [66].

### 4.2. Ferroptosis in Heart Failure

The development of heart failure is related to the loss of myocytes and is proven to have a relationship with ferroptosis. In an integrated bioinformatical analysis, Toll-like receptors 4 (TLR4) and NADPH oxidase4 (NOX4) were increased in HF. Knocking down TLR4 and NOX4 can remarkably inhibit myocyte death mediated by ferroptosis [12]. Another mechanism revealing the role of ferroptosis in HF is mixed lineage kinase 3 (MLK3). MLK3 induces ferroptosis by regulating JUN/p53 mediated oxidative stress in chronic HF. miR-351 can inhibit this kind of heart failure by suppressing the expression of MLK3 [77]. Iron is critical to ferroptosis, the homeostasis of which also plays an important role in HF. Disrupting the ferritin heavy chain (Fth) gene in mice leads to HF owing to iron deposition, whereas decreasing cardiac iron in mice by lacking transferrin receptor 1 (Tfr1) also leads to HF [78,79]. Furthermore, people suffering from HF also have decreased Tfr1 and cardiac iron [63].

There are several compounds to treat HF by regulating ferroptosis. Injecting TLR4-siRNA or NOX4-siRNA lentivirus can protect the failing heart via inhibiting ferroptosis. Puerarin, an antioxidant, can inhibit lipid oxidation and iron overload in mice with heart failure induced by isoprenaline [13]. Deferoxamine, a ferroptosis inhibitor, was reported to mitigate HF or cardiac infarction [15]. All the above suggests that ferroptosis might be a promising target for HF.

### 4.3. Ferroptosis in Cardiomyopathies

#### 4.3.1. Anticancer Drug-Induced Cardiomyopathy

Doxorubicin (DOX), a generally used anticancer drug, exerts severe side-effects, such as heart failure and cardiomyopathy. An RNA-sequencing result has shown that the expression of the heme oxygenase-1 (Hmox1) gene is significantly higher in mouse hearts treated with DOX than that of the control, which means Hmox1 is critical for DOX-induced cardiomyopathy [15]. Moreover, the role of Hmox1 is related to ferroptosis. Nrf2 can upregulate Hmox1 to degrade heme and release free irons, thus inducing ferroptosis in cardiomyocytes [14]. Although excess iron is hazardous, selectively cardiomyocyte Fth deficient mice also suffer mild cardiomyopathy, and dietary iron cannot rescue it but instead leads to severe heart injury through ferroptosis [63]; therefore, iron homeostasis is critical to cardiomyopathy related to ferroptosis. Furthermore, knocking down P53 can increase the expression of SLC7A11 to inhibit ferroptosis and administering Fer-1 is protective for DOX-treated animals [80,81]. Dexrazoxane (DXZ), one iron chelator, is an FDA-approved compound treating DOX-induced cardiomyopathy through inhibit ferroptosis. Mito-TEMPO, targeting mitochondria, the main organelle to produce DOX-Fe2+ induced lipid peroxide, suppresses DOX-related ferroptosis, and protects cardiomyocytes similar to Fer-1 [15,82].

#### 4.3.2. Diabetic Cardiomyopathy

Diabetes mellitus, closely related to heart structure and function damage, can lead to cardiomyopathy via ROS’s overproduction and reduction in antioxidant ability, which is an important promoting factor of ferroptosis. NRF2, inhibiting ferroptosis by modulating antioxidant expression to reduce lipid ROS, is a promising target for diabetic cardiomyopathy (DCM) [83,84]. Hydrogen sulfide (H2S) can increase glutathione to synthesize GSH in DCM, which reduces lipid peroxidation and further inhibits ferroptosis [85]. Consequently, ferroptosis is likely to play a role in DCM.

This section may be divided by subheadings. It should provide a concise and precise description of the experimental results, their interpretation, as well as the experimental conclusions that can be drawn.

#### 4.3.3. Iron Overload-Induced Cardiomyopathy

Iron overload (IO)-induced cardiomyopathy is thought to have a relationship with ferroptosis, which includes a high level of lipid oxidation and iron overload and usually occurs in hereditary hemochromatosis and diseases needing frequent blood transfusions such as thalassemia [86,87,88]. Treating cardiomyocytes with 80 μg/mL ferric ammonium citrate for 72 h promotes eicosanoids production and arachidonic acid release, which increases lipid peroxides and affects cardiomyocyte rhythmicity, suggesting a causal link between these signals and electromechanical abnormalities in IO-induced cardiomyopathy [89]. Mitochondrial Ca uniporter (mCU), which mediates iron uptake, is important to IO-induced cardiomyopathy. Compared to those of mCU+/+ (WT) mice, mCU−/− (KO) have lower mitochondrial iron and ROS levels and better systolic function; therefore, ferroptosis participates in IO-induced cardiomyopathy. Administering ferrostatin-1 protects heart function also confirms this point of view [90].

### 4.4. Ferroptosis in Atherosclerosis

Atherosclerosis is mainly caused by lipid deposition, endothelial cell damage, the proliferation of VSMC, and transformation of macrophages, and so on [54]. All these processes involve ferroptosis. Lipid deposition, especially PUFAs, is the substrate of the LOX or Fenton reaction, leading to oxidized lipids and inducing ferroptosis. In high-fat diet (HFD)-fed Apo E-/-mice, the administering of Fer-1, a ferroptosis inhibitor, can promote the expressions of SLC7A11 and GPX4, partially inhibit the iron accumulation and lipid peroxidation, as well as increase ox-LDL-induced mouse aortic endothelial cells (MAECs) viability [16]. Moreover, endothelial cells can be damaged by lipid ROS, increasing the endothelial permeability, promoting lipid deposition, and initiating atherosclerosis. In HUVECs, miRNA17-92 overexpression greatly alleviates erastin-induced cell death through the A20-ACSL4 axis [1]. That means inhibiting ferroptosis in endothelial cells can alleviate endothelial damage and reduce the probability of atherosclerosis. Studies showed that the iron level in healthy arterial tissue is less than that in atherosclerotic plaque [91]. Iron overload is also a key point to induce ferroptosis; therefore, restricting iron, especially nontransferrin bound iron (NTBI), a deleterious iron form, is thought to be therapeutic for atherosclerosis by inhibiting ferroptosis [54,92]. Furthermore, iron overload even changes macrophage function, leading it to polarize toward the proinflammatory subtype [58]. This chronic inflammation of the macrophage contributes to the formation of atherosclerotic necrotic core and destabilization of plaque [93]; therefore, ferroptosis can induce atherosclerosis via iron overload and macrophage changes.

### 4.5. Ferroptosis in Aging

Ferroptosis is related to aging because of the role of iron and excess ROS. Iron accumulates with aging as the need for iron decreases as the metabolic rate reduces, and the amount of hemoglobin also decreases with aging [94]. One experiment on rabbits shows that, compared to that of young ones, aged rabbits possess more free iron, released from ferritin, and this leads to further oxidative damage, which may be related to ferroptosis, because the ferroptosis inhibitor, deferoxamine, is protective of this kind of damage [95].

### 4.6. Ferroptosis in Vascular or Ventricular Remodeling

The homeostasis of iron is critical to ferroptosis and the pathogenesis of vascular and ventricular remodeling. One study shows that iron deficiency contributes to vascular remodeling in the pulmonary of rat [96]. On the contrary, another study reports that restricting iron may moderately alleviate hypoxia-induced vascular remodeling in the pulmonary of mice [97]. As for the heart, it is believed that the myocardial iron in the infarct zone is adverse for ventricular remodeling [98]; therefore, the role of iron and ferroptosis in vascular remodeling needs more investigation (Figure 2).

## 5. Pharmacologic Modulation of Ferroptosis in Cardiovascular Diseases

Several compounds could modulate ferroptosis; however, most of them are just applied in the experiment to test whether they are effective for cardiovascular disorders (Table 2). Of note, ferroptosis inhibitors seem to be dose-dependent. One study report that treating mice with either inhibitor at a lower dose (5 mg kg^−1^ compared with 10 mg kg^−1^) is also helpful but cannot reach the same extent [73]. Consequently, it is required to determine the appropriate dose for treatment.

## 6. Challenges at Present

Ferroptosis, an iron-dependent cell death, is reported to participate in the formation and regulation of cardiovascular diseases. Given that the causes and treatments of CVDs are complex, and ferroptosis can be a potential target for treatments, we can combine the pharmacologic modulation targeting of ferroptosis with the existing treatment to cope with the drug resistance.

Although the treatment is promising, most of the drugs are all limited to experimental stages. Furthermore, the treatment extent varies with the ferroptosis inhibitor dose, and the exact dose is uncertain. Moreover, because most of the CVDs are chronic, ferroptosis inhibitors should be applied for a long time. Besides, the drug distribution is not target-specific but all over the body, which may lead to several side effects. For example, deferoxamine, an iron chelator, is reported to be ototoxic and neurotoxicity [103]; therefore, safer and more effective formulations deserve more exploration.

There is also hidden trouble with ferroptosis inhibitors. It was demonstrated that ferroptotic inducers effectively inhibit cancers in many studies; therefore, when we use ferroptosis inhibitors for cardiovascular disease therapy, this may lead to carcinogenesis, which is a potential risk.

Interestingly, contradictory to our previous view that upregulating SLC7A11 is antiferroptotic, the expression of SLC7A11 was recently reported to have the opposite effect because the amount of cystine increased and led to a crisis of NADPH “debt” when reducing cystine to cysteine, which can impair the antioxidant ability and drive cells to ferroptosis [36].

Moreover, the role of energy stress in ferroptosis also creates controversy. Energy stress induces AMPK activation, leading to ACC inactivation and the decrease in PUFA and other fatty acids [51]; however, a study also reports that AMPK promotes ferroptosis by phosphorylating BECN1. Phosphorylated BECN1 can be bind to SLC7A11, then inhibits cystine transport [52].

Consequently, on account of the controversies we state above, more investigation is required to comprehensively decipher and detail the mechanism when targeting SLC7A11 and AMPK as treatments for CVDs.

## 7. Future Perspectives

Ferroptosis was demonstrated to be significant in cancer and thought to be a promising avenue for cancer therapy. Several clinical drugs were already proved to induce ferroptosis in tumor cells, such as sorafenib [104,105], sulfasalazine, salinomycin [106], artemisinin, and its derivatives [107,108,109,110]. Treatment of cancer-targeting ferroptosis is proved to be solid and bright. Given that ferroptosis also play an important role in CVDs, we can hope that the treatment of CVDs can reach the same extent as cancer soon by regulating ferroptosis.

Given that most CVDs are chronic, pharmacologic modulation of CVDs should be administered for a long time; however, some ferroptosis inhibitors such as deferoxamine are reported to be ototoxic and neurotoxic [103]; therefore, we should focus more on compounds with fewer side effects. Vitamin E, a natural compound, can inhibit ferroptosis by suppressing 15-LOX [101]. Moreover, one study has shown that feeding GPX4 knocking out mice with low vitamin E food leads to ferroptosis in endothelial cells, which shows that vitamin E is rather important to protect against ferroptosis [111]. Melatonin, an endogenous antioxidant, is also proved to inhibit ferroptosis induced by hemin, high glucose, traumatic brain injury, and hypoxic–ischemic brain damage [112,113,114,115]. These natural materials are safer and can be administered for a long time; however, the curative effect of ferroptosis needs more investigation and may be a hot topic of research in the future.

For a long time, ferroptosis was thought to be regulated only by iron metabolism, lipid peroxidation, and the Xc-GSH-GPX4 pathway; however, recently several novel regulatory pathways were proposed, such as FSP1–CoQ10–NAD(P)H axis, GCH1–BH4–DHFR pathway, and DHODH in mitochondrial. The regulatory pathways are comprehensively depicted in Figure 1. Moreover, NCOA4, Hmox1, BACH1, ACSL4, Nrf2, P53, CoQ10, GCH1 are key molecules in regulating ferroptosis, which were further studied in cardiovascular diseases. For example, in pressure-overloaded heart induced by transverse aortic constriction, NCOA4 was demonstrated to contribute to ventricular dilation, myocardial fibrosis, cardiac dysfunction, and hypertrophy, which is related to an increased content of ferrous ion induced by NCOA4-mediated ferritinophagy [116]. As for Hmox1, although it also increases ferrous iron by degrading heme, it exerts obvious protective effects and is a therapeutic target in CVDs, which is closely related to CO production in heme degradation [117]. Therefore, when focusing on the beneficial effects of Hmox1 in CVDs, we should also pay attention to the dark side of inducing ferroptosis. It was demonstrated that CoQ10 is an attractive treatment of CVDs [118,119,120]; however, we never considered that the mechanism of CoQ10 contains the effect of ferroptosis. Can we target CoQ10 to regulate ferroptosis and further treat CVDs? To what extent is the role of ferroptosis in the treatment of CoQ10? Are there any other downstream regulators of CoQ10 exerting therapeutic effect? In addition, regulatory molecules for ferroptosis such as DHODH and FSP1 have no relationship with CVDs yet; thus, will they become new targets for therapy? All these questions and other novel and effective regulatory mechanisms require further exploration.

Ferroptosis may not be an individual type of cell death. One study shows that the administering of a ferroptosis activator can increase autophagosomes. Moreover, NCOA4-mediated ferritinophagy is a kind of autophagy that leads to iron accumulation contributing to ferroptosis [121]. Although ferroptosis is defined as a new kind of cell death that is different from others, the relationship between ferroptosis and other kinds of cell death is vague. How can we regulate those kinds of cell death comprehensively? Whether ferroptosis is an independent type of cell death or just part of other existing types? Are there any other new pathways to regulate ferroptosis?

The existence of ferroptosis was demonstrated in CVDs by most existing studies, even though the research on ferroptosis and CVDs is still in its infancy. In fact, ferroptosis is characterized by redox imbalance and lipid peroxides accumulation, which participate in nearly all the pathogenesis of diseases related to ferroptosis. Therefore, reducing the source of lipid peroxide and increasing its metabolism seem to be the direction of therapy. Highly expressed in cardiomyocytes and catalyzing the formation of ROS, NOX4 are remarkably increased in heart failure. Knocking down TLR4 and NOX4 can remarkably inhibit myocyte death mediated by ferroptosis [12]. Antioxidant puerarin can inhibit lipid oxidation and iron overload in mice with heart failure [13]. As for I/R, ferroptosis inhibitor lip-1 protects heart I/R injury by reducing voltage-dependent anion channel 1 (VDAC1) levels and increasing GPX4 levels [10,11]. Iron homeostasis disturbance is also important to I/R [68]. Nrf2 is a promising target in cardiomyopathy, which is key in modulating antioxidant expression and reducing lipid ROS [83,84]. The iron level in healthy arterial tissue is less than that in atherosclerotic plaque [91]. Besides, increasing ox-LDL promotes the development of atherosclerosis. Restricting iron and ox-LDL is thought to be therapeutic for atherosclerosis by inhibiting ferroptosis [54,92]. With ongoing research, the mystery of ferroptosis will be further revealed. Inhibiting ferroptosis is likely to be a promising strategy for CVDs treatment.

Besides the diseases we mentioned above, hypertension, atrial fibrillation, deep vein thrombosis, and aortic aneurysm are critical in CVDs, but few studies illustrated their relationship with ferroptosis [122,123,124,125]. Are there any other CVDs that are linked to ferroptosis? How significant is this correlation? Can we prevent other CVDs by targeting ferroptosis? Although the significant role of ferroptosis in CVDs aroused intense scholarly interest and wide attention, all these questions merit further exploration.

## Figures and Tables

**Figure 1 biomolecules-12-00390-f001:**
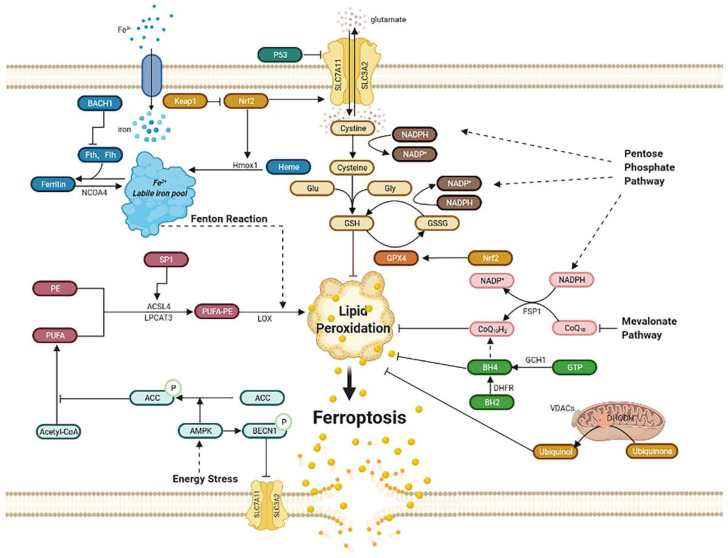
Regulatory pathways of ferroptosis. Ferroptosis is an iron-dependent, new kind of cell death. Regulatory pathways are complex, which can mainly be divided into several parts: iron metabolism; lipid peroxidation; Xc-GSH-GPX4 pathway; FSP1–CoQ10–NAD(P)H pathway; GCH1–BH4–DHFR pathway. Mitochondria and energy stress also play an important role. Abbreviations: BACH1: BTB domain and CNC homolog 1; Fth1: ferritin heavy chain 1; Ftl1: ferritin light chain 1; NCOA4: nuclear receptor coactivator 4; Hmox1: heme oxygenase-1; Keap1: Kelch-like ECH associated protein 1; Nrf2: nuclear factor erythroid 2-related factor 2; PE: phosphatidyl ethanolamine; PUFA: poly-unsaturated fatty acid; PUFA-PE: poly-unsaturated fatty acid-phosphatidyl ethanolamine; Sp1: special protein 1; ACSL4: acyl-CoA synthetase long-chain family member 4; LPCAT3: lysophosphatidylcholine acyltransferase 3; LOX: lipoxygenase; ACC: acetyl-CoA carboxylase; AMPK: AMP-activated protein kinase; BECN1: beclin 1; SLC7A11: subunit solute carrier family 7 member 11; SLC3A2: solute carrier family 3 member 2; Glu: glutamate; Gly: glycine; GSH: glutathione; GSSG: oxidized glutathione; GPX4: glutathione peroxidase 4; GCH1: guanosine triphosphate cyclohydrolase 1; GTP: guanosine triphosphate; BH4: tetrahydrobiopterin; BH2: dihydrobiopterin; DHFR: dihydrofolate reductase; DHODH: dihydroorotate dehydrogenase; VDACs: voltage-dependent anion channels; FSP1: ferroptosis suppressor protein1.

**Figure 2 biomolecules-12-00390-f002:**
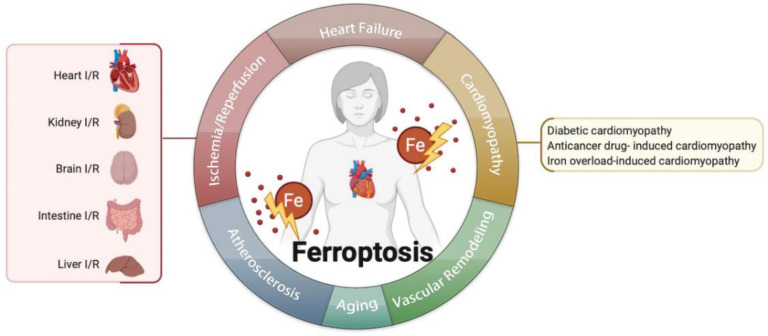
Ferroptosis is related to multiple cardiovascular diseases, such as ischemia/reperfusion, heart failure, cardiomyopathies, atherosclerosis, aging, vascular remodeling, and so on.

**Table 1 biomolecules-12-00390-t001:** Target ferroptosis to reduce I/R injury in different organs.

Organs	Models and Cells	Compounds	Targets	Mechanisms and Consequences	References
heart	Sprague–Dawley rats, H9c2 cell	Fer-1	ACSL4; GPX4	ACSL4 reduced; GPX4 increased; infarct size reduced.	[9]
heart	Mice (C57BL/6J)	Fer-1	AA	5-HETE, 11-HETE, 12-HETE, and 15-HETE reduced; infarct size reduced; left ventricular improved.	[8]
heart	Mice (C57BL/6J)	Lip-1	GPX4; ROS	GPX4 increased; ROS decreased; infarct size reduced; mitochondrial structural integrity and function maintained.	[10]
renal	HK-2 cells	ALR	GPX4; ROS	ROS decreased; ALR and GPX4 colocalized; renal I/R injury protected.	[67]
renal	Mice	2DG; AICAR; Fer-1	AMPK; PUFA	AMPK activation; renal I/R injury protected.	[51]
renal	Mice (C57BL/6) HK-2 cells	Panx1 deletion	Hmox1, NCOA4, Fth1	Hmox1 upregulated, NCOA4 and FTH1 inhibited; lipid peroxidation decreased; renal I/R injury protected.	[66]
renal	Mice (C57BL/6)	Fer-1		renal I/R injury protected.	[72]
intestine	Mice(C57BL/6) Caco-2 cells	Lip-1; ROSI	GPX4 ACSL4	GPX4 induced; ACSL4 inhibited; intestine I/R injury protected.	[34]
brain	Mice (C57BL/6) PC12 cells	LV-shRNA-PVT1 or LV-miR-214	GPX4, SLC7A11	GPX4 and SLC7A11 increased; infarct size reduced.	[69]
brain	Mice (C57BL/6)	Lip-1; Fer-1		infarct size reduced.	[73]
brain	Mice (Sv129/J) purified cortical neurons.	Desferrioxamine	HIF-1	HIF-1 increased; tolerance against reversible focal cerebral ischemia.	[74]
liver	Mice (C57BL/6)	Lip-1		liver I/R injury reduced.	[70]

**Table 2 biomolecules-12-00390-t002:** Pharmacologic compounds modulating ferroptosis in cardiovascular diseases.

Mechanisms	Compound	Model	Effects	References
reduce iron overload	deferoxamine	mice	mitigate HF or cardiac infarction.	[15]
	nanochelators	mice	the same as deferoxamine but have less side-effect and rapid renal excretion.	[99]
	desferrioxamine mesylate	human	improve left ventricular ejection fraction	[100]
	deferiprone	human	improve left ventricular ejection fraction	[100]
	deferasirox	human	improve left ventricular ejection fraction	[100]
	DXZ	mice	treating DOX-induced cardiomyopathy through inhibit ferroptosis	[15]
	rapamycin	mice	target mTOR to protect ferroptotic cardiomyocytes	[62]
reduce lipid ROS	Fer-1	mice	protect ferroptotic cardiomyocytes damage	[63]
	Fer-1	mice	promote the expressions of SLC7A11 and GPX4	[16]
	Fer-1	mice	alleviate DOX-induced cardiomyopathy	[80]
	Lip-1	mice	protect heart I/R injury by reducing VDAC1 levels and increasing GPX4 levels.	[10]
	vitamin E	mice	protect ferroptotic cell damage	[101]
	zileuton	mice	protect neurodegenerative disease by inhibiting 5-Lipoxygenase	[102]
	Mito-TEMPO	mice	suppress DOX related ferroptosis and protect	[15]

## Data Availability

Not applicable.

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
