# Peer review of "Ferroptosis in Cardiovascular Diseases: Current Status, Challenges, and Future Perspectives"

_biomolecules, 2022, doi:10.3390/biom12030390_

Round 1
Reviewer 1 Report
In this review article the authors have summarized various mechanisms of ferroptosis and discussed ferroptosis dependent development of cardiovascular diseases. Overall the manuscript is well written with appropriate sub heading and the manuscript is organized well. However, there are few topics that fall short of expectation and needs further discussion. Therefore following are my suggestion.
1) Please provide citation for the following statement “cardiovascular diseases (CVDs), a major cause of global mortality and disability, greatly affect people’s lives”.
2) Provide reference for each CVDs “Recent research has demonstrated that ferroptosis participates in several CVDs, such as I/R, heart failure (HF), cardiomyopathy, and atherosclerosis”
3)The regulating pathways of Ferroptosis “This concept, ferroptosis, was first proposed by Dixon,” - Start the sentence with “The concept or simply Ferroptosis”
4) “Overproduction of lipid peroxides, not clearing in time, affecting the normal structure and function of membrane, is the most” This sentence needs modification to properly convey the intended meaning
5)Overproduction of lipid peroxides, not clearing in time, affecting the normal structure and function of membrane, is the most direct cause of cell death. Iron metabolism is critical in the formation of lipid peroxides. Xc-GSH-GPX4 pathway, FSP1–CoQ10–NAD(P)H pathway, GCH1–BH4–DHFR pathway and mitochondria all play an important role in scavenging excess lipid peroxides and regulating the process of ferroptosis. – Please support the statements with appropriate references.
6) Rephrase the following sentence to denote what is both and use the word act of metabolize rather than using take “Both take poly-unsaturated fatty acid-phospha- tidyl ethanolamine (PUFA-PE) as substrates and produce lipid peroxides”
7)“Once the amount of lipid peroxides is out of control” what is out of control means rather change the sentence using words like abnormal or aberrant or hyperactive.
8)Provide proper citation for the following statement. “Consequently, increasing the expression or catalytic activity of ASCL4, LPCAT3, LOX, or Fenton reaction leads to lipid peroxides accumulation and, ultimately, ferroptosis”.
9) 2. 4. FSP1–CoQ10–NAD(P)H pathway- This section is too short and needs additional details.
10) 2.7. Energy stress – This topic lacks details and needs more discussion. Please refer to article PMID:35069910
11) Please include discussion on the article PMID: 35186185; PMID: 34988167 and PMID: 34760046 to “3.1. Endothelial cells “section.
12)Discuss the article PMID: 33513420 in section “3.2. Vascular smooth muscle cells”
13)Section on “3.3. Macrophages” is also underdiscussed and need to be elaborated. These articles may help PMID: 35110526 and PMID: 35036405
14) Include the following reference PMID: 32351005 to discussion on the topic “4.1. Ferroptosis in ischemia/reperfusion (I/R)”
15) Discussion on “4.3.3. Iron overload-induced cardiomyopathy” is short and the authors should elaborate more on this topic. Reference PMID: 11352890 can help.
16) Additional figure to demonstrate the signaling pathways involved in ferroptosis induced cardiovascular disease including heart failure , I/R and atherosclerosis and other vascular remodeling is warranted.
Reviewer 2 Report
The manuscript of Y. Guo et al. “Ferroptosis in cardiovascular diseases: Current Status, Challenges, and Future Perspectives” describes the molecular events participating in the development of ferroptosis, as well as the main signaling and metabolic pathways, and enzyme systems involved in its manifestation. The features of the development of ferroptosis in the cells of the vascular wall, cardiomyocytes and inflammatory cells are also considered. The possible role of ferroptosis is characterized in the development of myocardial ischemic-reperfusion injury, heart failure, and cardiomyopathy of different ethyology. The role of ferroptosis in atherosclerosis, aging and remodeling of the cardiovascular system is briefly characterized.
Based on the above information, the possibility of pharmacological correction of pathological conditions by affecting the components of the ferroptosis system is substantiated.
Reviewer has not major comments.
Minor comments, concerning designations and typographical errors in the text.
- It is desirable to decipher abbreviations of fig.1, otherwise it is difficult to understand the text of lines 48-51 and further. In the text, it’s also recommended to refer to fig.1 where it is relevant. In the figure, the arrows indicating NADP+--NADPH transition should be arranged correctly corresponding to cysteine-cystine and GSSG-GSH coupling.
- DHODH (lines 151, 155 and so on), the authors never decipher this abbreviation.
- Line 356 and following, table 2: Deferoxamine or desferoxamine?!
- Line 382, …which can be bind…
